# Nasopharyngeal Carcinoma Burden and Its Attributable Risk Factors in China: Estimates and Forecasts from 1990 to 2050

**DOI:** 10.3390/ijerph20042926

**Published:** 2023-02-08

**Authors:** Ruhao Zhang, Yifei He, Bincai Wei, Yongbo Lu, Jingya Zhang, Ning Zhang, Rongxin He, Hao Xue, Bin Zhu

**Affiliations:** 1School of Public Health and Emergency Management, Southern University of Science and Technology, Shenzhen 518055, China; 2School of Public Policy and Administration, Xi’an Jiaotong University, Xi’an 710049, China; 3Vanke School of Public Health, Tsinghua University, Haidian District, Beijing 100084, China; 4Freeman Spogli Institute for International Studies, Stanford University, Stanford, CA 94305, USA

**Keywords:** nasopharyngeal carcinoma, joinpoint regression, GBD study, risk factors, age-period-cohort model, prediction, burden of diseases

## Abstract

Nasopharyngeal carcinoma (NPC) is an uncommon and aggressive malignant head and neck cancer, which is highly prevalent in southern and southwestern provinces in China. The aim of this study was to examine the disease burden and risk factors of nasopharyngeal carcinoma in China from 1990 to 2019 and to predict the incidence trends from 2020 to 2049. All data were extracted from the 2019 Global Burden of Disease (GBD) study. Joinpoint regression and age-period-cohort (APC) models were chosen to analyze prevalence trends. The temporal trends and age distribution of risk factors were also analyzed descriptively. Bayesian APC models were used to predict the prevalence from 2020 to 2049. The results indicate a higher disease burden in men and older adults. Their attributable risk factors are smoking, occupational exposure to formaldehyde, and alcohol use. We predict that the incidence will be on the rise in all age groups between 2020 and 2049, with the highest incidence in people aged 70 to 89 years. In 2049, the incidence rate is expected to reach 13.39 per 100,000 (50–54 years), 16.43 (55–59 years), 17.26 (60–64 years), 18.02 (65–69 years), 18.55 (70–74 years), 18.39 (75–79 years), 19.95 (80–84 years), 23.07 (85–89 years), 13.70 (90–94 years), and 6.68 (95+ years). The findings of this study might deserve consideration in China’s NPC prevention and control policy design.

## 1. Introduction

Nasopharyngeal carcinoma (NPC) is an uncommon and aggressive malignant head and neck cancer deriving from the nasopharynx epithelium [1]. Tumors usually appear in the pharyngeal recess [2]. Early symptoms of NPC are mild and atypical, which can be manifested as pain in the nose, ears, neck, or head, so it is diagnosed in the middle and late stages [3]. Mortality after metastasis of nasopharyngeal carcinoma is extremely high [4].

Some risk factors are related to NPC. The HLA gene located in the MHC region of the 6p21 chromosome is a genetic susceptibility gene that increases the risk of nasopharyngeal carcinoma [5]. In addition to genetic factors, EBV infection is probably the most common cause of nasopharyngeal carcinoma [6]. Activation of EBV is necessary in the pathogenesis of nasopharyngeal carcinoma, and malignant transformation of EBV-infected epithelial cells requires support from other factors [7]. Poor eating habits can also increase the risk of nasopharyngeal carcinoma, such as eating salted fish [8], pickled vegetables, salted eggs, shrimp sauce, and bacon, etc. [9], and insufficient fruit and vegetable intake [10]. In addition, occupational exposure and environmental pollution, such as exposure to wood chips [11], formaldehyde [12], etc., are also risk factors for nasopharyngeal carcinoma. At the same time, smoking [13], high BMI, diabetes [14], less brushing of teeth, and more fillings will also increase the risk of nasopharyngeal carcinoma.

Nasopharyngeal carcinoma occurs worldwide, but there are significant geographical disparities in the prevalence of nasopharyngeal carcinoma [15]. The incidence of NPC in China and Southeast Asian countries is high, followed by North Africa, and the incidence rates in Europe, the United States, and the Pacific are less than one per 100,000 [16]. Statistics released by the International Agency for Research on Cancer show that in 2020, there were about 133,354 new cases of nasopharyngeal carcinoma in the world, and China is the country with the highest incidence of nasopharyngeal carcinoma, with about 60,000 newly diagnosed cases, accounting for 46.8% of the world. The age-standardized incidence rate (ASIR) of NPC in China was 3.0 per 100,000 in 2018 [2]. The incidence is particularly high in southern Chinese cities. One of the reasons for the high incidence of nasopharyngeal cancer in southern China may be that people there are more likely to consume certain food (such as salted fish) than people in other countries and regions [8]. As the incidence and mortality rate of NPC gradually increases with age, and China’s aging population is increasing, the burden of the disease will continue to grow. It can be seen that the prevention and control situation is serious. A global analysis shows that the incidence and mortality rates of nasopharyngeal carcinoma have gradually declined in recent decades [17], but they are increasing in China. As a result, nasopharyngeal carcinoma remains a major public health problem in China, which faces a heavy burden of disease.

The academic world has paid attention to the burden of NPC in China. In the past, some scholars have studied the incidence and mortality of nasopharyngeal carcinoma in a specific year in China, as well as the changing trend of the disease burden of nasopharyngeal carcinoma in China in the past. However, few researchers have paid attention to the prediction of the NPC burden and its related risk factors in China.

Therefore, we conduct this study to explore the incidence, mortality, disability-adjusted life years (DALYs), risk factors, and temporal and age trends of NPC in China from 1990 to 2019. Moreover, we perform the age-period-cohort analysis for different genders, and we also predict the trend of incidence of Chinese NPC in the next 30 years. The results above would provide valuable evidence for the intervention, treatment, and public health policy formulation of NPC in China.

## 2. Materials and Methods

### 2.1. Data Source

The Crude Incidence Rate (CIR), Crude Death Rate (CDR), Crude DALY rate, Age Standardized Incidence Rate (ASIR), Age Standardized Death Rate (ASDR), and age-standardized DALY rate related to this cancer were searched by sex and age in the global burden of disease 2019 (GBD 2019) database (https://vizhub.healthdata.org/gbd-results/) (accessed on 1 July 2022) from 1990 to 2019. Data on the three main risk factors (smoking, occupational exposure to formaldehyde, and alcohol use) for ASDR and age-standardized DALYs were collected by sex and age group. The GBD 2019 database, developed by the Institute for Health Metrics and Evaluation (IHME) at the University of Washington, synthesizes the available evidence on levels and trends in health outcomes, a range of different risk factors, and health system responses, and it shows us the burden of different types of diseases all over the world.

Matrix data of the population were collected from the United Nations Population Division’s World Population Prospects (2019 Revision) (https://population.un.org/wpp/Download/Standard/CSV/) (accessed on 1 July 2022) for morbidity projections. This report collects and predicts the total population of different countries and regions in the world from 1950 to 2100.

### 2.2. Statistical Analysis

We compared the trends of NPC burden in China from 1990 to 2019 through the data of incidence, deaths, and DALYs of different sex. To show the time trends more clearly, we used the joinpoint regression method, which can easily evaluate trends in the estimated annual percentage change. In the joinpoint regression model, all possible joinpoint points were established by searching, and the corresponding error sum of squares and mean square error in each possible case were calculated. The grid points with the smallest MSE were selected as joinpoint points, and piecewise regression was established according to the selected connection points and interval function fitting equation parameters according to the time characteristics of disease distribution. Under the condition of Poisson distribution, the software uses the rate to fit the log-linear model. The annual percent change (APC) was calculated by this model to examine the changes in ASIR, ASDR, and age-standardized DALY rate. Furthermore, the direction and magnitude of change were also determined by calculating the average APC (AAPC) and corresponding 95% confidence intervals (Cis).

We performed a descriptive analysis of temporal and age trends in risk factors for nasopharyngeal carcinoma in China. GBD 2019 reports the effects of smoking, occupational exposure to formaldehyde, and alcohol consumption on ASDR and age-standardized DALY rates for the disease in China. Therefore, we analyzed temporal trends in the contribution of these four risk factors from 1990 to 2019. In China, smoking and alcohol use were more closely linked to the ASDR and age-standardized DALYs of NPC than occupational exposure to formaldehyde. Therefore, in 2019, our focus was on the age difference between these two risk factors.

Age effect is the change associated with physiological and social processes to an individual’s aging. The period effect is the result of external factors affecting all age groups equally at a particular time. The cohort effect is the change caused by a unique experience over time. These three effects are an obvious collinearity. Age-period-cohort (APC) models fit the effects of age, period, and cohort as factors, and have always been used to find statistics on the rate information of death or incidence of a disease. Based on age-period-cohort analyses, we evaluated the effects of age, period, and birth cohort on time trends, splitting the temporal variations into the 3 components. Firstly, we divided the data into 10 age groups (50–54, 55–59, 60–64, 65–69, 70–74, 75–79, 80–84, 85–89, 90–94, 95+ years). Then, in the whole observation period, we divided the period into 6 groups (1990–1994, 1995–1999, 2000–2004, 2005–2009, 2010–2014 and 2015–2019). Finally, 15 birth cohorts (1892–1896, 1897–1901, …, 1957–1961, 1962–1966) were obtained by age and period. We estimated age, period, and cohort effects using the natural logarithm of disease incidence as the dependent variable and the median of these datasets as the independent variable, respectively.

Based on Bayesian age-period-cohort (APC) models, dividing the population group into 10 age groups (50–54, 55–59, …, 85–89, 90–94, 95+ years), we predicted the nasopharyngeal carcinoma incidence rate in China from 2020 to 2049 in different age groups.

### 2.3. Software

We calculated APC (annual percent change) and AAPC (the average annual percent change) of ASIR, ASDR, and age-standardized DALY rate of NPC by using the Joinpoint Regression Program (version 4.9.1.0). The Stata Program (version 16.0) was used to analyze the age-period-cohort model of NPC. Bayesian APC model was used by the package Nordpred in R (version 4.2.1) to predict the incidence rate of NPC in the next 30 years. All figures were drawn using OriginPro 2021 (version 9.8.0.200).

## 3. Results

### 3.1. Nasopharyngeal Carcinoma Burden in China

Figure 1 shows that males had a higher burden of NPC in terms of incidence, death, and DALY rate than females during 1990–2019 in general. For 30 years, the incidence rate of NPC had been on the rise in total. Totally, the CIR of NPC increased from 2.71 (per 100,000) to 7.76 during the 30 years. In terms of gender, the CIR of males increased by 234.21%, which was higher than the 102.45% growth rate of females. After the age-standardized, though the ASIR was still in an upward trend, the upward trend decreased significantly. The ASIR had increased 70.89%, compared to the 186.63% increase rate of CIR of NPC from 1990 to 2019. For the same situation, the ASIR of males had increased by 97.78%, and the increase rate of females was 21.35%.

Different from the upward trend of CIR and ASIR, the CDR and Crude DALY rate was on a slight downward trend. For all groups, the CDR and Crude DALY rate of NPC had decreased by 10.24% and 18.86%, respectively, from 1990 to 2019. Among males, the CDR had increased by 3.12%, and the Crude DALY rate had decreased by 6.6% on the contrary. Among females, the downward trend was obvious; their CDR and Crude DALY rates had decreased by 34.26% and 42.04%. Though there was an increasing trend of ASIR of NPC, the downward trend was evident in ASDR and age-standardized DALY rate. The ASDR and age-standardized DALY rates had decreased by 50.79% and 50.26%, respectively. By gender, the decline of ASDR and age-standardized DALY rate was more pronounced in females than in males. The ASDR had decreased by 43.11% for males and 64.37% for females. Paying attention to the age-standardized DALY rate, we found that males had decreased to 42.17% and females had decreased to 64.81%.

### 3.2. Joinpoint Regression on Time Trends of Nasopharyngeal Carcinoma Burden Analysis

Table 1 presents the time trends of NPC from 1990 to 2019. The AAPC result of ASIR shows that the incidence rate of NPC in China was on an upward trend (AAPC = 1.804). However, there was no significant upward trend from 1996 to 2006 (*p*-value > 0.05). On average, ASIR of females went up 0.689% per year and ASIR of males went up 2.326% per year. Among females, during 1996–1999 and 2010–2013, the changing trend of ASIR was not significant. For males, there was no obvious upward trend in ASIR from 1999 to 2003. The changing trend was down at first (1990–1996), and then the ASIR increased.

The AAPC result of ASDR shows that the death rate of NPC in China was on a downward trend (AAPC = −2.454). The ASDR of females decreased by 3.498% per year and the ASDR of males decreased by 1.958% per year (all *p*-value < 0.05). At the same time, the DALY rate of NPC in China was in a downward trend as well (AAPC = −2.394). The age-standardized DALY rate of females decreased by 3.536% per year and the age-standardized DALY rate of males decreased by 1.898% per year. Except for males, the changing trend of the DALY rate during 2014–2019 was not obvious (*p*-value > 0.05); all the others were significantly decreased (*p*-value < 0.05).

### 3.3. Impact of Different Risk Factors on Nasopharyngeal Carcinoma Burden

In Figure 2, we analyzed the impact of three different risk factors (smoking, occupational exposure to formaldehyde, and alcohol use) on NPC. In 1990, the influence factors of smoking, occupational exposure to formaldehyde, and alcohol use attributing to death were 0.7374, 0.0209, and 0.9537, respectively, and the influence factors of age-standardized DALYs were 21.03, 0.8953, and 30.87, respectively. When times went to 2019, the influence factors of these three risk factors for death were 0.4264, 0.0117, and 0.5852, respectively, and the influence factors of age-standardized DALYs were 12.34, 0.5287, and 19.45, respectively. Among the three risk factors, smoking and alcohol use had a heavier burden compared to occupational exposure to formaldehyde based on the ASDR and age-standardized DALY data. During the 30 years, all the influence factors for these three risk factors first decreased rapidly and then tended to be stable. Additionally, in all three risk factors, the ASDR and age-standardized DALY rates of males were higher than those of females.

Figure 3 presents the difference in ASDR and age-standardized DALY rates for NPC by smoking and alcohol use across age groups. We found that both rates showed a trend of increasing first and then decreasing with the increase in age groups. From the effects of smoking and alcohol use on ASDR of NPC, ASDR reached its peak in the 85–89 age group and then started to decrease rapidly when over 90 years old. Alcohol use was a greater risk factor for ASDR on NPC than smoking in the 30–79 and 90+ age groups. In terms of the influence of smoking and alcohol use on the age-standardized DALY rate of NPC, alcohol use was most significant in the 60–64 age group. Furthermore, the age-standardized DALY rate of smoking to NPC was most significant in the 65–69 age group.

### 3.4. APC Model Analysis of Nasopharyngeal Carcinoma in China

Figure 4 and Table 2 show that the age effect on NPC showed an increase first, and decrease over 90 years old, along with the time of both sexes in China. On the contrary, the period effect and the cohort effect on males showed a steady trend, and effects on females showed a decline at first and an increase after reaching the nadir. Additionally, the three kinds of effect coefficients in females were more obvious along with time. For people between 50 and 69 years, the age effect coefficient for males was larger than for females. When the age hits 85–89 years, the age effect coefficient for females decreased rapidly, while the trend for males was less significant. For the period effect, the trend for males was less pronounced (the minimum value was −0.113 and the maximum value was 0.258). However, before 2007, the period effect coefficient for females decreased from 0.899 (1992) to −0.681 (2007). Additionally, the period effect coefficient increased after 2007; the value was up to 0.192 in 2017. The cohort effect coefficient for females declined rapidly in the birth cohorts of 1892–1911 and 1922–1941. After the birth cohort of 1942, the cohort effect coefficient for females increased rapidly. Furthermore, before the birth cohort of 1911 and after the birth cohort of 1952, the cohort effect coefficient for females was larger than for males.

### 3.5. Prediction of Nasopharyngeal Carcinoma Incidence Rate in Different Age Groups

Based on the Bayesian APC model, we predicted the NPC incidence in China in different age groups. From Figure 5, we found that the incidence rate was on an upward trend in every age group. With the increase of age, and the changing trend of the incidence rate of the NPC group of 70 to 89 years old, the trend of NPC incidence rate fluctuated greatly. At the same time, this age group has the highest incidence rate of NPC.

Accurate across all age groups, in 2049, the incidence rate is expected to reach 13.39 per 100,000 (50–54 years), 16.43 (55–59 years), 17.26 (60–64 years), 18.02 (65–69 years), 18.55 (70–74 years), 18.39 (75–79 years), 19.95 (80–84 years), 23.07 (85–89 years), 13.70 (90–94 years), and 6.68 (95+ years). As the age group increases, the incidence increases slowly at first and then decreases rapidly after 90 years old.

## 4. Discussion

Based on GBD 2019 data, we analyzed the disease burden of nasopharyngeal carcinoma and its related risk factors in China from 1990 to 2019, and predicted the incidence rate of NPC in different age groups from 2020 to 2049. The results showed that the CDR, ASDR, Crude DALY rate, and age-standardized DALY rate decreased from 1990 to 2019. However, the CIR and ASIR of NPC increased. According to the forecast results of different age groups, the incidence tends to rise in all age groups in 2020–2049, with the highest incidence in the 70–89 age group.

Our findings suggest that both CIR and ASIR for NPC in China showed an upward trend from 1990 to 2019, which may be related to increased exposure to risk factors for NPC. Smoking [13] is one of the important risk factors for NPC. China accounts for 30% of the world’s smoking population [18]. According to the China Smoking Hazards To Health Report 2020 issued by the National Health Commission, the number of smokers in China exceeded 300 million, and the smoking rate of people over 15 years old in China in 2018 was 26.6%. Drinking [19] is another important risk factor for NPC. The Scientific Research Report on Dietary Guidelines for Chinese Residents (2021) shows that the drinking rate is 64.5% for men and 23.1% for women, of which the smoking rate for men was 50.5% and that for women was 11.8 million. In addition, occupational exposure to formaldehyde [20] is also a risk factor for NPC. Formaldehyde is an occupational carcinogen present in many different working environments. Workers in industrial processes (resin, plastics, textile, leather, rubber, cement, and plastics industries), professionals in anatomical and pathological laboratories, etc., are among those at high risk of exposure to formaldehyde [21].

However, despite the increasing CIRs and ASIRs for both sexes, the ASDRs for both sexes and CDR for females is decreasing. This may be due to the availability of early screening [22] and advances in treatment [23]. Early screening methods for NPC, such as EBV antibody detection [22], can enable patients to be diagnosed in the early stage of cancer. Patients with early-stage (stage I and II) NPC have a good treatment effect, and the five-year survival rate is as high as 94% [24]. The treatment of NPC has improved due to advances in radiotherapy techniques, and induction and concurrent chemotherapy [25]. Five-year survival rates for patients with NPC have improved.

In terms of gender differences, although the ASDRs for both sexes decreased, CDR for males increased. It can be seen that men are at higher risk than women. This difference may be due to the greater exposure of men to risk factors such as smoking, alcohol, industrial pollutants, and more.

A global analysis [2] shows that the incidence of nasopharyngeal carcinoma has gradually declined in recent decades, while the situation in China runs counter to global trends, showing a trend of increasing year by year, and will continue to rise in the next three decades. The increase in incidence may be related to the promotion of early screening. The global trend of nasopharyngeal cancer mortality is decreasing. In most countries and regions (Singapore, Hong Kong, Israel, etc., in Asia; South Africa; the United States and Canada in North America; and most of Europe), the mortality rate for nasopharyngeal cancer is declining, with APCs reaching −3.1% to −4.3% in the countries with the greatest declines. Nasopharyngeal cancer mortality in China is also on the decline [26,27], which may be due to the improvement of medical standards [7], lifestyle changes [28], and the reduction of exposure risk factors [24].

According to the forecast results of different age groups, the incidence tends to rise in all age groups in 2020–2049, with the highest incidence in the 70–89 age group. We found that older adults have a higher risk of disease than younger people. The higher incidence in older people may be due to age; older people over the age of 60 are at a high risk of tumors, with cancers such as lung, stomach, liver, esophagus, and colorectal cancers being highly prevalent in the elderly population. Possible reasons for this are: tumor formation is usually a long process; the older a person is, the longer he or she is exposed to carcinogenic factors; and the reduced immunity of older people. The incidence of both lung [29] and gastric cancers [30] increases with age. Gastric cancer is uncommon in adults under the age of 45, and the median age of diagnosis is 70 years [31]. The incidence of esophageal cancer rises rapidly with age, with the average age at diagnosis being 67 years [32]. Older adults have higher mortality rates probably because their poorer health conditions reduce the effectiveness of treatment [33]. Another reason why the disease burden of the elderly group is higher than that of the young may be the delay in seeking medical treatment [34]. Cancer is often in the middle and advanced stages when it is diagnosed [3], and if patients delay seeking medical treatment, the condition will deteriorate further. In addition, it is estimated that around 2035, China’s elderly population aged 60 and above will exceed 400 million, entering a stage of severe aging. This will make the disease burden of NPC a major public health problem. Therefore, in areas of China with a high incidence of nasopharyngeal carcinoma, prevention and control can be carried out in the following ways. For occupational exposure to formaldehyde as a risk factor, good prevention and control of occupational diseases should be promoted, such as: employers should take appropriate protective measures for materials and equipment used that may generate occupational exposure hazards; and occupational health guardianship files should be established for workers and kept properly for a certain period of time. In addition, screening programs for nasopharyngeal carcinoma should be developed for key areas and populations. Health education should be strengthened to improve the health literacy of the whole population, and publicity and education efforts should be stepped up to enable the public to accept correct concepts and health knowledge and to form good living behaviors.

This study had certain limitations. We analyzed the prevalence trend of NPC in China through the GBD 2019 and predicted its incidence over the next 30 years. Although the GBD 2019 consolidates a large amount of data, there may still be some omissions of data from remote and poor areas. In addition, we analyzed the overall disease burden of NPC in China; however, due to a lack of provincial data, we were unable to compare the differences in disease burden in different regions of China. Furthermore, the GBD 2019 does not account for the role of risk factors such as EBV infection and dietary habits. We did not perform a sensitivity analysis, which may have caused limitations in the prediction results.

## 5. Conclusions

The disease burden of nasopharyngeal carcinoma in China is generally on the rise. Compared with young people, the disease burden of the elderly is heavier, mainly because increasing age leads to the accumulation of exposure to risk factors, and their poorer health conditions reduce the effectiveness of treatment. Men have a higher burden of disease compared to women, which may be due to their greater exposure to harmful environments such as tobacco, alcohol and formaldehyde. The incidence rate tends to rise in all age groups in 2020–2049, with the highest incidence in people aged 70 to 89 years. Therefore, it is important to prevent and control occupational exposure, develop screening programs for nasopharyngeal cancer for key areas and populations, and strengthen health education.

## Figures and Tables

**Figure 1 ijerph-20-02926-f001:**
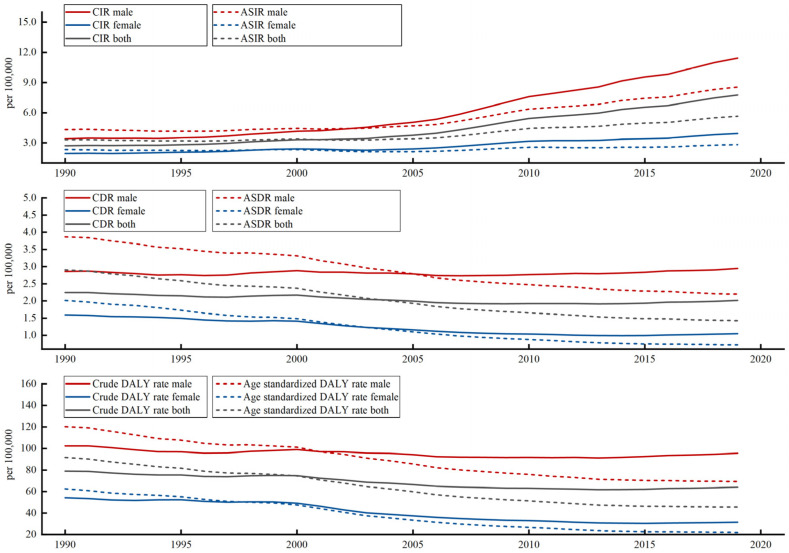
Trends of musculoskeletal disorders burden in China from 1990 to 2019.

**Figure 2 ijerph-20-02926-f002:**
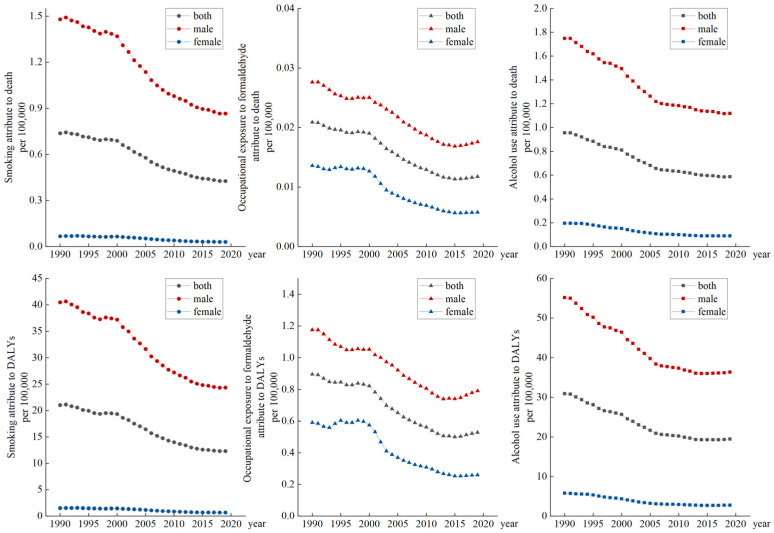
The variation trends of ASDR and age-standardized DALY rates of three risk factors in different genders over 30 years.

**Figure 3 ijerph-20-02926-f003:**
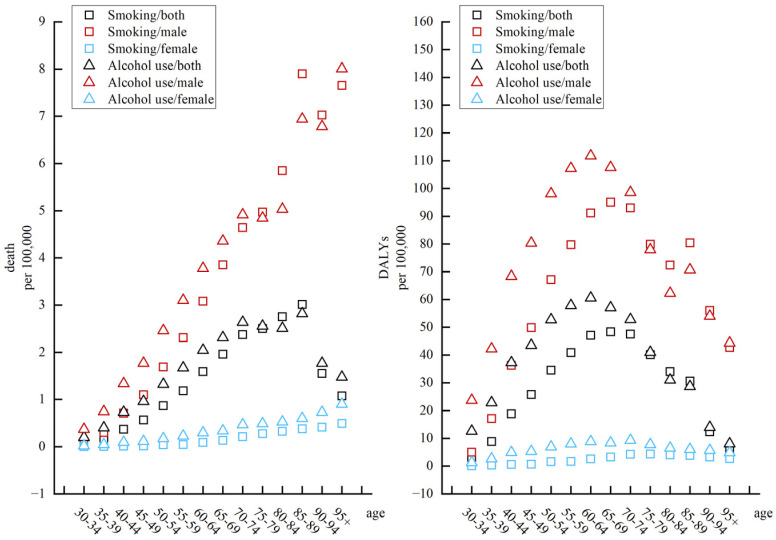
The variation trends of ASDR and age-standardized DALY rates of two risk factors in different genders and age groups in 2019.

**Figure 4 ijerph-20-02926-f004:**
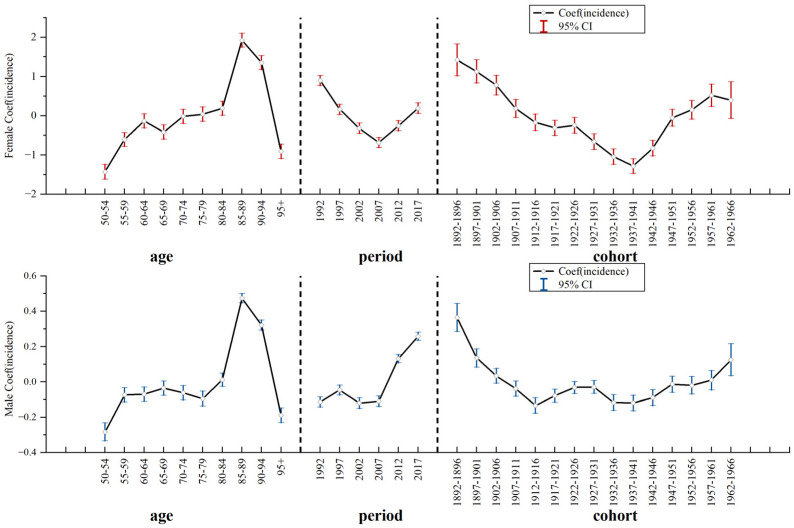
APC model analysis of nasopharyngeal carcinoma incidence among females and males in China.

**Figure 5 ijerph-20-02926-f005:**
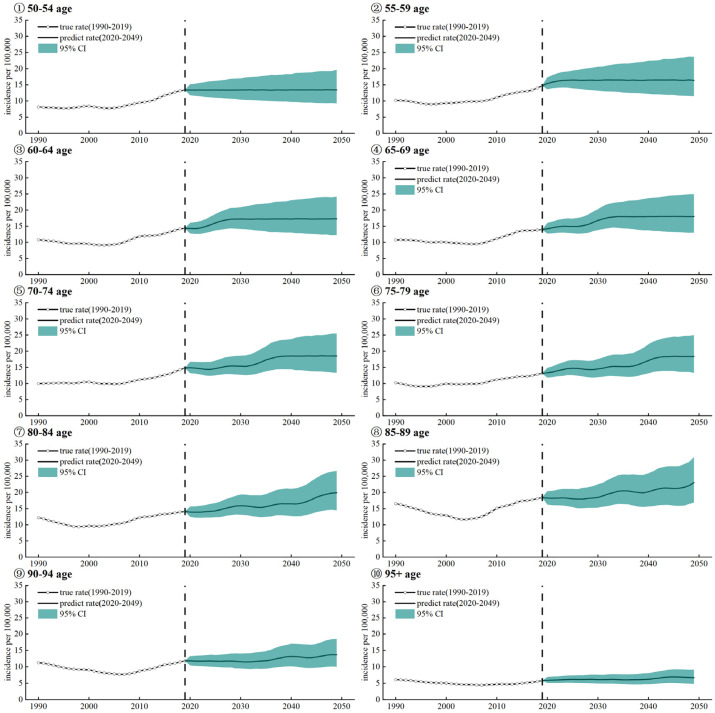
Prediction of nasopharyngeal carcinoma incidence among females and males in China from 2020 to 2049 based on the Bayesian APC model.

**Table 1 ijerph-20-02926-t001:** Log-transformed joinpoint trends of Nasopharyngeal carcinoma ASRs by sex in China.

Measure	Sex	Trend 1		Trend 2		Trend 3		Trend 4		Trend 5		Trend 6		1900–2019 AAPC (95% CI)
		Years	APC	Years	APC	Years	APC	Years	APC	Years	APC	Years	APC	
Age-standardized incidence Rate	Both	1990–1996	−0.825 *	1996–1999	2.105	1999–2003	−0.598	2003–2006	2.033	2006–2009	6.701 *	2009–2019	2.786 *	1.804 (1.296–2.315)
	Female	1990–1996	−0.760 *	1996–1999	1.879	1999–2005	−1.918 *	2005–2010	4.166 *	2010–2013	−0.793	2013–2019	1.861 *	0.639 (0.131–1.149)
	Male	1990–1996	−0.778 *	1996–1999	2.018 *	1999–2003	0.247	2003–2006	2.954 *	2006–2010	6.825 *	2010–2019	3.303 *	2.326 (1.982–2.671)
Age-standardized death Rate	Both	1990–1997	−2.484 *	1997–2000	−1.273 *	2000–2007	−4.052 *	2007–2014	−2.270 *	2014–2019	−1.107 *	NA	NA	−2.454 (−2.578–−2.329)
	Female	1990–1994	−2.686 *	1994–1997	−4.433 *	1997–2000	−2.144 *	2000–2007	−5.744 *	2007–2014	−3.568 *	2014–2019	−1.078 *	−3.498 (−3.671–−3.325)
	Male	1990–1996	−1.993 *	1996–2000	−1.034 *	2000–2007	−3.417 *	2007–2014	−1.595 *	2014–2019	−1.097 *	NA	NA	−1.958 (−2.076–−1.84)
Age-standardized DALY Rate	Both	1990–1997	−2.430 *	1997–2000	−1.172	2000–2003	−4.704 *	2003–2007	−3.944 *	2007–2014	−2.360 *	2014–2019	−0.446	−2.394 (−2.522–−2.265)
	Female	1990–2000	−2.649 *	2000–2003	−7.583 *	2003–2007	−5.695 *	2007–2014	−3.672 *	2014–2019	−0.844 *	NA	NA	−3.536 (−3.788–−3.282)
	Male	1990–1996	−2.393 *	1996–2000	−0.903 *	2000–2007	−3.276 *	2007–2014	−1.777 *	2014–2019	−0.311	NA	NA	−1.898 (−2.017–−1.779)

Notes: AAPC, Average annual percent change; APC, Annual percent change; CI, confidence interval; NA, not applicable. * Significantly different from zero, *p* value < 0.05.

**Table 2 ijerph-20-02926-t002:** APC model analysis of nasopharyngeal carcinoma incidence among females and males in China.

Incidence	Female		Male	
	Coef.	*p* > z	Coef.	*p* > z
Age (years)				
50–54	−1.429 (−1.618, −1.239)	0.000	−0.283 (−0.334, −0.232)	0.000
55–59	−0.604 (−0.779, −0.429)	0.000	−0.074 (−0.113, −0.034)	0.000
60–64	−0.134 (−0.312, 0.044)	0.141	−0.069 (−0.109, −0.029)	0.001
65–69	−0.417 (−0.597, −0.236)	0.000	−0.035 (−0.075, 0.004)	0.078
70–74	−0.016 (−0.197, 0.164)	0.859	−0.062 (−0.103, −0.021)	0.003
75–79	0.04 (−0.141, 0.221)	0.663	−0.095 (−0.137, −0.053)	0.000
80–84	0.192 (0.012, 0.373)	0.037	0.012 (−0.026, −0.05)	0.531
85–89	1.922 (1.744, 2.101)	0.000	0.474 (0.448, 0.501)	0.000
90–94	1.353 (1.177, 1.529)	0.000	0.321 (0.293, 0.349)	0.000
95+	−0.908 (−1.092, −0.725)	0.000	−0.189 (−0.23, −0.148)	0.000
Period (year)				
1992	0.899 (0.771, 1.027)	0.000	−0.113 (−0.142, −0.085)	0.000
1997	0.161 (0.03, 0.293)	0.016	−0.046 (−0.074, −0.019)	0.001
2002	−0.318 (−0.45, −0.187)	0.000	−0.12 (−0.15, −0.091)	0.000
2007	−0.681 (−0.812, −0.55)	0.000	−0.109 (−0.139, −0.08)	0.000
2012	−0.253 (−0.382, −0.125)	0.000	0.132 (0.108,0.155)	0.000
2017	0.192 (0.056, 0.329)	0.006	0.258 (0.235, 0.28)	0.000
Cohort (year)				
1892–1896	1.42 (1.018, 1.823)	0.000	0.364 (0.285, 0.444)	0.000
1897–1901	1.128 (0.833, 1.423)	0.000	0.136 (0.084, 0.187)	0.000
1902–1906	0.782 (0.531, 1.034)	0.000	0.034 (−0.008, 0.076)	0.111
1907–1911	0.184 (−0.042, 0.411)	0.111	−0.038 (−0.081, 0.005)	0.086
1912–1916	−0.168 (−0.377, 0.041)	0.115	−0.134 (−0.177, −0.091)	0.000
1917–1921	−0.309 (−0.501, −0.116)	0.002	−0.078 (−0.115, −0.041)	0.000
1922–1926	−0.246 (−0.445, −0.048)	0.015	−0.032 (−0.066, 0.002)	0.068
1927–1931	−0.659 (−0.858, −0.459)	0.000	−0.029 (−0.065, 0.007)	0.113
1932–1936	−1.044 (−1.239, −0.848)	0.000	−0.117 (−0.161, −0.073)	0.000
1937–1941	−1.281 (−1.468, −1.095)	0.000	−0.119 (−0.164, −0.075)	0.000
1942–1946	−0.825 (−1.024, −0.626)	0.000	−0.089 (−0.134, −0.044)	0.000
1947–1951	−0.054 (−0.268, 0.161)	0.624	−0.013 (−0.059, 0.032)	0.561
1952–1956	0.152 (−0.087, 0.39)	0.212	−0.02 (−0.069, 0.03)	0.431
1957–1961	0.52 (0.237, 0.803)	0.000	0.01 (−0.046, 0.065)	0.729
1962–1966	0.399 (−0.07, 0.868)	0.095	0.125 (0.034, 0.216)	0.007
Constance	6.816 (6.742, 6.89)	0.000	2.818 (2.803, 2.832)	0.000

Notes: APC model, age–period–cohort model; Coef, coefficient; and CI, confidence interval.

## Data Availability

The data used in this study are openly available in GBD 2019 at https://vizhub.healthdata.org/gbd-results/ (accessed on 12 August 2022).

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
