# Peer review of "Nasopharyngeal Carcinoma Burden and Its Attributable Risk Factors in China: Estimates and Forecasts from 1990 to 2050"

_ijerph, 2023, doi:10.3390/ijerph20042926_

Round 1

Reviewer 1 Report

#The authors should described global NPC mortality rate is a downward trend. Would you add more detail about it in discussion. Why NPC patient rate is increasing in China? 

#Why does NPC has significant geographical disparities? Does risk factors exposure rate in Chinese and Southeast Asians have different from other countries?

#What is the situation for occupational exposure to formaldehyde. Please give us more detail.

#In page 12, line 298,  Higher incidence in older people may be due to age, as increasing age, . . . . .. Please add more explanation that the almost malignant disease, incidence rate may be increasing in older people. Since, it is not only NPC but also other malignant disease.

#The authors described that men are at higher risk than women for prevalence trend of NPC. This may be due to men's lifestyle and career choices being different from women's in Discussion. It is politically incorrect. The world will not accept this line of thinking.

#What should we do to decrease incidence and mortality rate of NPC. Please add it in Discussion and Conclusion.

#How about EBV infection is one of the risk factor? EBV is one of the most common infection in the World people. Would you describe the authors thinking.

Reviewer 2 Report

Regarding the contents of China's nasopharyngeal cancer, I think it could be a good report on the fact that the risk of occurrence by age group was analyzed. However, if data are included on how the disease differs from other countries in terms of cultural differences in each country, especially smoking, alcohol, or occupational characteristics, the disease is increasing, and in particular, the age group in their 50s, which has the most risk factors, is included. It's a pity that it wouldn't have been a more universal study.
